# Conducting 24-Hour Dietary Recalls in Group Settings with Adults Having Low-Income: Perspectives of EFNEP Peer Educators

**DOI:** 10.3390/nu15184020

**Published:** 2023-09-17

**Authors:** Karen Franck, Michael Puglisi, Annie J. Roe, Susan Baker, Teresa Henson, Dawn Earnesty, Kavitha Sankavaram

**Affiliations:** 1Department of Family and Consumer Sciences, The University of Tennessee, Knoxville, TN 37996-4501, USA; 2Department of Nutritional Sciences, University of Connecticut, Storrs, CT 06269, USA; michael.puglisi@uconn.edu; 3Margaret Ritchie School of Family and Consumer Sciences, University of Idaho, Moscow, ID 83844-3183, USA; aroe@uidaho.edu; 4Department of Food Science and Human Nutrition, Colorado State University, Fort Collins, CO, USA; susan.baker@colostate.edu; 5Family and Consumer Sciences Department, University of Arkansas at Pine Bluff, Pine Bluff, AR 71601, USA; hensont@uapb.edu; 6Health and Nutrition Institute, Michigan State University Extension, Saginaw, MI 48607, USA; wilcoxd4@msu.edu; 7Department of Nutrition and Food Science, University of Maryland, College Park, MD 20742, USA; kavitha@umd.edu

**Keywords:** nutrition education, low-income, dietary measures

## Abstract

The Expanded Food and Nutrition Education Program (EFNEP) is a federally funded program that teaches nutrition education to adults and youth with low-income. EFNEP is funded throughout the United States including federal territories. The purpose of EFNEP is to provide nutrition education. Evaluation for adult programs includes pre/post surveys and pre/post 24-h diet recalls (24HDR). A validated standard of dietary measures, 24HDR are useful when collected as designed: one-on-one by a trained professional. In EFNEP, 24HDR are collected in group settings by EFNEP peer educators who often have not received a college degree or any formal education in nutrition. The purpose of this study was to explore attitudes and behaviors of EFNEP peer educators regarding how they collect diet recalls in a group setting, their perceptions of how adult participants feel about the recalls, and the benefits and challenges of using recalls. Online interviews were conducted with EFNEP peer educators across the U.S. Peer educators recognized the importance of collecting the recall data but identified several challenges such as time, resources, and participant reluctance to complete the recall. Program evaluation through methods like the 24HDR is important to measure outcomes and inform program improvements but also needs to include how evaluation can benefit participants and minimize data collection burden. Future research needs to examine the validity of collecting recalls in a group setting compared to other measures of diet quality.

## 1. Introduction

The Expanded Food and Nutrition Education Program (EFNEP) is a community outreach program that aims to reduce nutrition insecurity of low-income families and youth [1]. The program is federally funded by the U.S. Department of Agriculture and National Institute of Food and Agriculture (USDA/NIFA). EFNEP is the nation’s first nutrition education program and operates through the 1862 and 1890 Land-Grant Universities (LGUs) in every state, district and territory [1]. 

Since 1969, EFNEP has been delivered by peer educators, who are recruited and hired from the communities EFNEP serves. Typically, job requirements for peer educators are a high school diploma or General Educational Development (GED) certification. Peer educators recruit families and receive referrals from community organizations and agencies. Peer educators are trained, supervised, and supported by universities and locally based professionals. 

For the past two decades, EFNEP has had a history of rigorous evaluation for adult programs that includes a standardized, validated pre/post questionnaire [2]. In addition to the questionnaire, adult participants complete a 24-h dietary recall (24HDR) before and after the EFNEP lesson series. The pre/post 24HDR responses are used to assess initial eating habits, potential nutrient deficiencies, and any changes in eating behaviors that may result from the program. 

Administered properly, 24HDR is an effective and accurate assessment of dietary intake. One study compared 24HDR with self-recorded food diaries for 3653 participants. The estimated proportion of calories from fat, carbohydrates, alcohol, and protein reported in the 24HDR differed by 2.4% from the food diary, and the consumption between the two methods varied in three of ten food groups indicating that the 24HDR is an accurate method of dietary intake [3]. Previous research has identified best practices to increase accuracy of 24HDR that includes having a trained professional collect the recall using a multiple-pass method with standardized probes to elicit details about foods consumed [4]. 

The results of the 24HDR are utilized to show local, state and national impact from the EFNEP program. Annual data aggregation is used to show improvements in participants’ diets and nutrition practices.

Research has focused on the delivery of the 24HDR in an individual setting as opposed to the current group education program delivery model for EFNEP. The purpose of this research is to determine the use and efficacy of collecting 24HDR information in group settings from adults enrolled in EFNEP classes from the perspectives of the EFNEP peer educators. EFNEP peer educators are the frontline workers who interact with EFNEP participants directly. It is therefore critical to identify the ways of collecting complete and accurate data from EFNEP participants and to understand the challenges peer educators face in collecting 24HDR.

## 2. Materials and Methods

This qualitative study was part of a larger mixed methods study intended to answer questions about the use and efficacy of collecting 24HDR information from adults enrolled in EFNEP classes. EFNEP has evolved from providing one-on-one nutrition education in a participant home to group education in a community organization setting. Since COVID-19 there has been an increase in virtual group classes within the EFNEP program. This study focused on in-person educational program delivery. Pre and post 24HDR are typically administered in a group setting and are collected by EFNEP peer educators as a part of the educational time. In general, during class time EFNEP peer educators provide a form for each participant to complete and the peer educator provides verbal instructions to guide the participants through the 24HDR form. The form includes instructions for participants to include information about what they ate at different times (first meal of the day, snacks, etc.), portion sizes (e.g., cups and ounces), and details about condiments and how food was prepared (e.g., fried or baked chicken). These paper forms are entered into a national database for EFNEP that calculates the 24HDR information. Data entry into this system is either performed by the peer educators or by a data entry person depending on the institution. 

### 2.1. Study Design

Online, semi-structured interviews with EFNEP peer educators were conducted. A purposive sampling method was used to include peer educators that represented different levels of funding and different regions of the country. There are 76 EFNEP programs, each overseen by a program coordinator. At the federal level, EFNEP is divided into 4 geographical regions: North Central, North East, South, and West. The program coordinators in these regions have regular meetings to provide updates and share ideas. 

Researchers first asked EFNEP coordinators to nominate peer educators as part of an online survey about the 24HDR. Program coordinators were asked to provide names and contact information for EFNEP peer educators who had worked at least for 1 year with the program and worked at least 50% with adult audiences. Because nominations from the survey did not include peer educators from every region, the researchers also emailed specific EFNEP coordinators and regional listservs in order to recruit peer educators that represented all regions. 

This study was approved by the University of Maryland, College Park, Institutional Review Board (1416811-1). EFNEP peer educators completed an online consent form before interviews were conducted. 

### 2.2. Study Sample

Researchers selected educators to contact about the interviews based on region and type of institution in order to include diverse perspectives. Thirty interviews were conducted with EFNEP peer educators who were employed by 25 different LGUs or 33% of the 76 LGUs who receive EFNEP funding (Table 1).

### 2.3. Data Collection

The researchers emailed nominated peer educators with information about the project and asked them to participate in an online interview that would last one to two hours. The email included a link to an online consent form and potential dates for the interviews. The researchers scheduled online interviews using Zoom with peer educators who responded with availability and completed the online consent survey.

The researchers developed a script and pilot tested the script with 3 peer educators. Slight modifications were made to the questions based on the pilot tests and those interviews are included in this sample. Questions and probes included 6 main domains: perceptions and perceived benefits of the 24HDR, process of the 24HDR, training, challenges, and strategies (Table 2). Background and demographic questions were also asked at the beginning of the interview.

One to two researchers conducted the interviews (SB, KF, MP, AR, KS). When two researchers were present, one researcher was the primary interviewer and the other researcher took notes and asked follow-up questions if needed. Interviews were recorded using Zoom. 

### 2.4. Data Analysis

This qualitative study applied a modified grounded theory approach. This approach includes open coding that allows coders to organically identify themes and relevant codes rather than using established codes [5]. Interviews were transcribed through Zoom. Recordings were used to clarify transcripts that were not clear or had errors.

One researcher (KF) worked with a trained assistant to analyze the transcripts. Transcripts were organized using EXCEL. Using inductive coding, the researcher and the assistant separately identified themes and coded each interview. The two coders met to discuss themes and codes and come to consensus on any discrepancies. All discrepancies were resolved through discussion and transcript review.

## 3. Results

### 3.1. Description of the Sample

Almost all of the participants were women. Most were experienced with the years of employment with EFNEP ranging from a low of 3 years to a high of 34 years (mean = 11.53, SD = 7.26). Interviews were conducted with peer educators who represented each region in 2021 and 2022 (Table 1).

Peer educators were asked to describe their own feelings and attitudes about collecting 24HDR from adult participants with low-income, as part of the EFNEP data collection process. They were also asked to describe their perception of how participants felt about the 24HDR process. There were 4 main themes that emerged: challenging paperwork, time consuming process, intrusive process, and perceived benefits. 

### 3.2. Challenging Paperwork

Peer educators made several comments about the challenges they faced explaining the 24HDR to participants. Most peer educators (66.7%) reported that they collected the 24HDR in the first class—they believed that this contributed to the difficulty of engaging participants in a tedious and personal process before trust had been established. By design, the 24HDR is a repetitive process in order to help participants accurately include all foods and details that contribute to dietary intake. However, peer educators felt like the process could be overwhelming for participants. 

“It’s kind of tedious for the person who has to fill it out … with the listing of when they ate, how it was prepared, portion sizes, because sometimes they don’t really know how to explain portion sizes.”(peer educator 24)

Some peer educators commented that the 24HDR process might be easier if this was collected after participants had attended several classes and were more comfortable and trusting. For example, one peer educator recommended collecting the 24HDR during the third class. Others noted that the exit 24HDR (collected at the last class) seemed to be more accurate because participants understood how the 24HDR could be helpful to improve behaviors.

“Once you build that relationship, clients are more likely to feel better about opening up about what they’ve been eating.”(peer educator 2)

“[At exit 24HDR], either [participants] are more comfortable with me at that point or they have made some behavior changes they are really excited about.”(peer educator 15)

The 24HDR forms were another challenge identified by peer educators. This included the instructions on the forms that some peer educators felt participants might misunderstand or find confusing. 

“They’re overwhelmed because they think they have to fill up every line on all the lines on both sides, so that can be confusing to people.”(peer educator 21)

Almost half of the peer educators (46.7%) reported that they were teaching classes with adults who spoke English as a second language so several comments about the 24HDR forms were specific to these populations. For example, one peer educator noted that the serving sizes on the forms were English measures (e.g., cups and ounces) but many of the adults she worked with were familiar with the metric system so they had a difficult time completing serving sizes. Another noted that they do not always have forms in the language of origin for people to complete. Furthermore, language and cultural barriers extended to concepts on the forms such as snacks. One talked about working with translators and not being certain that the translator was following the 24HDR recall script that the peer educator was using.

Peer educators also identified literacy issues for participants. This included both being able to read the forms and embarrassment about spelling and writing out foods.

“It’s a whole lot for people that read even less than an 8th grade level.”(peer educator 19)

“The participants still can’t fill [24HDR form] out the way it’s supposed to be filled out. Just getting them to write their name on each piece of paper can be hard sometimes.”(peer educator 14)

### 3.3. Time Consuming Process

The process also was time consuming both for the participants and for the peer educators. Half of the peer educators reported that they were responsible for data entry as well as data collection so entering the diet recall into the online reporting system was also time consuming. 

Some peer educators described frustration about how the time participants spent completing forms took away from class time that they felt would be better spent helping participants learn nutrition information and skills to improve behaviors. Peer educators made comments about how it was difficult to have enough time for participants to complete the 24HDR forms along with other required paperwork as well as to provide an engaging activity for the first session so that participants would want to come back for the next lesson. Some peer educators questioned if the 24HDR process resulted in participants dropping out of the program. 

“Everyone’s gracious, and they don’t seem to have a real issue with it. But I do notice that people leave and don’t come back, and I always wonder is it because of the food recall and all of the paperwork we’re asking or did something just come up.”(peer educator 5)

“I think [24HDR] is one of the deterrents of getting people to stay the six to eight weeks with us because then they know they’re going to do that [24HDR] again.”(peer educator 7)

“You don’t have time to do a program if you’re going to get a real thorough 24HDR. You can devote a whole hour to do that—to get it really good—but you’re going to lose your people.”(peer educator 22)

### 3.4. Intrusive Process

Some peer educators identified the process as intrusive and personal for participants. Comments included perceptions that participants were embarrassed to include foods that they normally eat as well as participants who worried that they might have not had a typical day the day before. This included atypical days where participants might have overeaten certain foods like holidays or days where participants might have skipped meals because of illness or because they did not have enough money to buy food.

“Most of the time they are okay with [24HDR] because I tell them this is vitally important for me to conduct the program. However, I get things like ‘This is too much information to have to share.”(peer educator 15)

“Participants will often say ‘Yesterday wasn’t a normal day for me’ so [24HDR] doesn’t reflect how they typically eat.”(peer educator 18)

Other comments included the idea that participants might not provide accurate information because of social desirability. Peer educators described feeling that participants were worried that their diets would be judged by the educator.

“Many don’t mind providing it. They’re just more concerned about what I’m going to say about what they eat.”(peer educator 12)

“Some of them in the room—I can see panic on their face. They are worried that they didn’t eat well or can’t remember what they ate.”(peer educator 1)

### 3.5. Perceived Benefits

Some peer educators identified the benefits of the 24HDR process for participants. They talked about how it helped participants to recognize areas where they could change their eating habits such as cutting back on sugar-sweetened beverages or eating more fruits and vegetables. Some peer educators appreciated having information that they could share with their participants that described where they were doing well and what could be improved. 

“I like getting to know … what they are eating. That’s really helpful to have a 24-h recall to say ‘Here’s some things we could change or do differently.”(peer educator 30)

“I think it’s a great concept. I truly, truly do especially since it gives a person an opportunity to evaluate what they’re actually taking in. And then to be able to give them more information about how to add to it, how to enhance it, or perhaps even how to prepare it in a better way.”(peer educator 24)

## 4. Discussion

A primary focus of EFNEP is to improve dietary intake of adults with low-income. In order to assess program impact on dietary intake, it is necessary to have an effective method of data collection that provides accurate information without undue burden to the participants and peer educators. As a national program, EFNEP’s methods of data collection also need to be standardized across states and territories. In this qualitative study, the perspectives of EFNEP peer educators were captured. While peer educators acknowledged important benefits of conducting 24HDR for EFNEP participants, they also noted that challenging paperwork and the time consuming and intrusive process made conducting 24HDR difficult in EFNEP. 

A longstanding strength of EFNEP is to improve dietary behaviors, and peer educators in this study often mentioned the benefits related to being able to utilize the 24HDR to display areas where participants could improve and areas where participants were making healthy choices. A recent study by Fuller et al. [6] echoed this sentiment, with EFNEP participants indicating a greater awareness of their dietary intake as a result of the 24HDR, which is a strategy that has been shown to be effective for improving dietary intake [7]. Additionally, some peer educators in this study expressed an appreciation to adjust and tailor their approach based on the 24HDR data to meet the needs of their participants.

The 24HDR has been validated for providing accurate data in one-on-one settings with trained professionals [8]. Gills et al. [9] reported that peer educators can also obtain accurate dietary data via this one-on-one method when trained appropriately. However, EFNEP peer educators often conduct 24HDR in group settings, a method that has not been validated. Additionally, previous research has indicated that 24HDR are often less accurate for populations with low educational attainment [10] and overweight and obese populations [11], aligning with a large percentage of the EFNEP audience [12,13]. Furthermore, it is not well established that a one-day 24HDR is as effective at capturing diet quality as repeated measures that include non-consecutive weekdays and weekends [14]. This research is needed particularly with EFNEP adults who are from low-resource households where food availability may change on a daily basis. In this current study, peer educators shared concerns about the accuracy of 24HDR with EFNEP adults especially conducting recalls in group settings with constraints such as time and literacy.

Many peer educators in this study shared that participants felt that the 24HDR process was too personal, and might result in participants not wanting to provide accurate information about their dietary intake that would be viewed negatively. DeBiasse et al. [15] reported similar findings for conducting dietary assessment for low-income women, emphasizing a need to remove personal questions and ensure that the process is culturally appropriate. 

A major theme of the interviews was that the 24HDR in EFNEP is a very time-consuming process, creating a burden for the participants and peer educators. Retaining participants for the entire EFNEP lesson series has been a challenge across the US [16], and peer educators often felt that the amount of paperwork and time required by the 24HDR may cause some participants not to return for subsequent lessons. EFNEP participants interviewed by Fuller et al. [6] also discussed the high burden of completing the 24HDR, related to the high cognitive load associated with remembering details related to their intake. 

Group collection of 24HDR takes more time to conduct, and peer educators in this study indicated that larger groups take considerably longer, a difficulty in EFNEP also reported by Townsend et al. [13]. Participants may have limited literacy and English language proficiency, further adding to the time-consuming nature of the process of obtaining specific dietary intake details. Additionally, many peer educators reported that they must also enter the 24HDR into the online system used for EFNEP. Peer educators expressed frustration with the time it takes to complete data entry, especially when adding cultural foods that are not in the system’s database. Townsend et al. [13] also cited this difficulty with accurate data entry for 24HDR in EFNEP related to system limitations. 

The excessive time required to complete the group 24HDR in EFNEP was also previously discussed by Blackburn et al. [17]. Researchers compared the process of collecting data from EFNEP participants via the 24HDR to a validated food behavior checklist, and reported that the checklist saved a considerable amount of time, allowed for easier data entry and detected the behavioral changes resulting from the EFNEP class series. Along with the 24HDR, a validated questionnaire [2] is also required to be administered before and after the EFNEP series. This tool reflects changes in dietary behaviors as a result of EFNEP classes and could provide similar benefits to the checklist in Blackburn et al. [17], serving as a less time consuming, more accurate and more manageable method to assess dietary behavior change.

At a programmatic level, it is important to periodically review program protocols and data collection procedures to assess utility, benefits, and administration burden for participants as well as alignment with program goals and objectives. The 24HDR is a valid measure of diet behaviors when administered by a trained professional using a 5-pass method in a one-on-one interview [18]. However, in EFNEP, typically the 24HDR is administered in a group setting by peer educators. As found in this study, EFNEP peer educators identified the process as being time-intensive and intrusive resulting in potential barriers to engage and retain adult participants in the program. 

EFNEP provides important nutrition education for adults with low-income. As with any program, there are limitations on potential impacts on behavior because of funding. EFNEP classes are short-term, reliant on collaborations with community organizations who provide space and time for classes, and on participants who voluntarily attend the classes in most circumstances. Successful EFNEP peer educators must be able to build trusting relationships with adult participants [19]. If the 24HDR is to be continued in EFNEP, one simple recommendation to help improve participant engagement would be to collect the 24HDR in the second class in an effort to reduce the amount of paperwork collected at the beginning of the program. Colorado State University has incorporated this structure into their EFNEP curriculum and could serve as a model for other curricula [20]. 

Another recommendation is to assess whether the benefits of 24HDR results equal or exceed the time and funding invested in data collection and analyses. Other benefits of the 24HDR need to be assessed given the high level of administrative burden. Limitations of 24HDR include biases related to social desirability and underreporting [18,21,22]. These biases can be reduced when the 5-pass method is followed but it is not clear that this method translates to a group setting in terms of accuracy [17]. With all of these associated costs in time and funding, it is clear that there needs to be compelling evidence for the benefits of the 24HDR results both for local and national programs and for program participants. This study did not show that peer educators are consistently using 24HDR results to inform and tailor programs locally so that would be one area that would need to be strengthened. 

EFNEP is the oldest national nutrition education program for low-income audiences and has been funded and implemented since the 1960s when it was created as part of President Lyndon Johnson’s War on Poverty. As described in a historical review by Leidenfrost [23], initially EFNEP was implemented by indigenous peer educators from low-income communities who provided nutrition education to individual mothers. The 24HDR and a questionnaire were introduced in 1976 to measure and monitor program outcomes, and over the years EFNEP shifted from individual education to educating groups of low-income adults in community settings in order to increase reach [23]. Since 1976, the adult questionnaire has changed significantly and the current questionnaire includes validated questions that assess changes in all five of EFNEP’s priority areas: diet quality, food resource management, food safety, food security, and physical activity [1]. Given that the current adult questionnaire includes validated questions that assess changes in key dietary and physical activity behaviors, it is important to assess the benefits of continuing to use the 24HDR with EFNEP adult participants. This is especially true since the questionnaire takes less time to administer, all dietary behavior questions have been tested and validated with EFNEP adult audiences, and the results can be aggregated nationally with more confidence and accuracy than the data available from the 24HDR. As EFNEP has adapted to meet the changing needs for their audiences, it is important that the program measures continue to change and adapt as well. 

## Figures and Tables

**Table 1 nutrients-15-04020-t001:** Characteristics of peer educators.

Characteristic	*n* (%)
Sex Female	29 (97)
Male	1 (3)
Location	
North Central	4 (13)
North East	10 (33)
South	9 (30)
West	7 (23)
Race	
African American/Black	12 (40)
White	15 (50)
Not identified	3 (10)
Ethnicity	
Hispanic/Latino	3 (10)
Not Hispanic/Latino	27 (90)

**Table 2 nutrients-15-04020-t002:** Domains and question examples.

Domain	Question Examples
Perceptions and perceived benefits of the 24HDR	What comes to your mind when you think of 24HDR? In your opinion, what do you think your participants think about providing the 24HDR information?
Process of 24HDR	When you are collecting 24HDR, is it usually done in a group setting or with individual participants? How do you explain or introduce the 24HDR process to your participants?
Training	Have you received specific training on how to collect 24HDR? If yes, was this training helpful when collecting data from your participants?
Challenges	What are your biggest challenges when collecting 24HDR? What would help you with these challenges?
Strategies	What are some suggestions that you think would help other educators get accurate 24HDR information? What do you think would help new less experienced educators be better prepared to collect 24HDR?

## Data Availability

The data presented in this study are available on request from the corresponding author.

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
