# Peer review of "Conducting 24-Hour Dietary Recalls in Group Settings with Adults Having Low-Income: Perspectives of EFNEP Peer Educators"

_nutrients, 2023, doi:10.3390/nu15184020_

Round 1

Reviewer 1 Report

The manuscript presents a captivating exploration of the EFNEP, shedding light on its essential role in educating individuals with low-income about healthy nutrition practices. The author masterfully delves into the program's objectives, highlighting its nationwide impact and significance. The manuscript underscores the importance of the program's evaluation, particularly through the use of pre/post surveys and the gold-standard pre/post 24-hour diet recalls (24HDR). The unique challenge of conducting 24HDR in group settings by EFNEP peer educators is adeptly discussed, offering valuable insights into the program's operational intricacies.

The manuscript's engaging narrative artfully brings to the forefront the attitudes and behaviors of EFNEP peer educators, unveiling their perceptions and experiences in collecting diet recalls. The study deftly navigates the challenges faced by these educators, including time constraints, resource limitations, and participant reluctance. By conducting online interviews with EFNEP peer educators from across the U.S., the manuscript skillfully captures their perspectives, thereby enriching the discourse on the practicalities of recall collection.

Furthermore, the manuscript thoughtfully suggests that program evaluation not only measures outcomes but should also consider the participants' benefits and the reduction of data collection burden. This perspective reflects a commendable understanding of the multifaceted nature of program assessment.

In conclusion, this manuscript stands as a captivating and insightful exploration of the EFNEP program and its challenges, effectively highlighting the dedication of peer educators and the potential implications for program improvement. The author's engaging presentation and deep understanding of the subject matter make it a valuable contribution to the field of nutrition education.

Author Response

Thank you for this review. We appreciate the feedback.

Reviewer 2 Report

Thank you for providing the opportunity to review this manuscript. The qualitative study, authored by a research team from various institutions, delves into the perceptions of peer educators concerning the advantages and challenges of conducting 24-hour recalls in a group setting within low-income communities. I commend the researchers for addressing this pivotal topic. Below are some suggestions to enhance the manuscript:

Abstract:

Line 16: Consider revising to "provides nutrition education" or simply "teaches nutrition."

Lines 22-24: It might be clearer to specify the purpose by adding "in a group setting," especially if contrasting with one-on-one methods.

Line 27: Could you specify what the "program evaluation" refers to?

Introduction:

Lines 60-61: Which is more accurate: the 24 HDR or the food diary?

Lines 69-71: Similar to the abstract, it might be beneficial to indicate that this study focuses on group settings.

Methods:

Line 79: Could you provide a citation for the mentioned theory? A brief introduction to the theory might also be beneficial.

Concerning data analysis: Did you utilize any specific software for coding? Was any triangulation method applied to resolve discrepancies between coders?

Discussion:

When discussing findings that align with previous research, it would be valuable to further delineate how your study adds to the existing literature.

Conclusion:

Lines 320-322: Can you provide a citation here?

Lines 334-343: This section seems detailed for a conclusion. Consider streamlining.

Lines 351-357: This information might be more suitably integrated with related content in the introduction.

I trust these suggestions will be constructive in refining the manuscript. Again, I appreciate the effort and research behind this work.

Author Response

Thank you for your feedback. Please see the Word file that includes specific feedback for each of your concerns.

Reviewer 3 Report

Thank you for the opportunity to review the manuscript "Conducting 24-Hour Dietary Recalls in Group Settings with Adults Having Low-Income: Perspectives of EFNEP Peer Educators". Overall, an interesting read however there are a few methodological and reporting flaws that should be addressed. Please see my comments below:

- Lines 40-56 are more methods on the project background rather than highlighting why this research is needed and could be moved to the methods section

-Method section 2.2 - could you please add more about how these 30 participants were chosen in this section 

- in the methods could you please add information on how 24hr recalls are usually collected by educators? Is it the same method across all areas? Do you use programs or are they manually transcribed. Do you use the same databases? Some more information on how these are collected would help provide insights into your project, particularly as a key finding is that the process is time consuming. Therefore it would be good to add more about this in the methods.

- Were any other tools besides Zoom used to transcribe and interpret interviews? 

- Were there any qualitative methodologies employed for this analysis? What kind of approach was used to identify themes? More detail is needed

- Could you please provide some examples of the questions asked under each domain?

- Lines 130-133 are more methods than results

- Could the quantitative data also be included in the summary table e.g. "Most peer educators (66.7%) reported that they collected the 137 24HDR in the first class"

-There are a lot of quotes used in the results, it reads like you have just added all quotes in. Could you limit to 1-2 key quotes? E.g. Section 3.4 - there are 7 quotes

- Line 250 - is this mentioned in the results at all? I don't recall that it was - it would be good to ensure this is mentioned in the results if you are discussing it

- The conclusion is very long and almost a whole other discussion. Could the conclusion title be moved to the end with an overall summary. 

Author Response

Thank you for your feedback. Please refer to the attached Word file that includes answers to your specific concerns.

Round 2

Reviewer 3 Report

Thank you for updating the manuscript and providing more comprehensive methods. This manuscript has been greatly improved.

- line 19 could you please rephrase this sentence, 24hr recalls are reliant on memory and are not the gold standard dietary assessment. Perhaps you could say validated or traditional. 

- line 332 has a note that says needs citation that should be removed 

Author Response

Thank you for these comments. 

Line 19 has been rephrased.

Line 332 has been corrected.
